# T-MLP: Tailed Multi-Layer Perceptron for Level-of-Detail Signal Representation

## Abstract

Level-of-detail (LoD) representation is critical for efficiently modeling and transmitting various types of signals, such as images and 3D shapes. In this work, we propose a novel network architecture that enables LoD signal representation. Our approach builds on a modified Multi-Layer Perceptron (MLP), which inherently operates at a single scale and thus lacks native LoD support. Specifically, we introduce the Tailed Multi-Layer Perceptron (T-MLP), which extends the MLP by attaching an output branch, also called *tail*, to each hidden layer. Each tail refines the residual between the current prediction and the ground-truth signal, so that the accumulated outputs across layers correspond to the target signals at different LoDs, enabling multi-scale modeling with supervision from only a single-resolution signal. Extensive experiments demonstrate that our T-MLP outperforms existing neural LoD baselines across diverse signal representation tasks.

## 1 Introduction

Representing signals with neural networks is an active research direction, known as implicit neural representation (INR) (Sun et al., 2022; Molaei et al., 2023; Essakine et al., 2024). Unlike traditional discrete signal representation that stores signal values on a fixed-size grid, INR represents a continuous mapping from coordinates to signal values using a neural network, offering a more compact representation than conventional discrete grid-based representations. Moreover, due to the smooth nature of neural networks, INR allows for the straightforward computation of derivatives of the signal. These advantages have propelled active studies in using INR for representing various types of signals, such as images (Chen et al., 2021; Skorokhodov et al., 2021; He & Jin, 2024), videos (Sitzmann et al., 2020; Fathony et al., 2021; Yan et al., 2024), and 3D shapes (Park et al., 2019; Gropp et al., 2020; Chabra et al., 2020; Wang et al., 2023; Yang et al., 2025).

Most INRs are based on Multi-Layer Perceptrons (MLPs), which operate at a single scale and lack support for multiple levels of detail (LoDs). Specifically, an MLP requires all of its parameters to be available in order to produce meaningful outputs; for instance, an MLP with $N$ hidden layers cannot function properly if only the parameters of the first $N - 1$ layers are available. Thus, those INRs based on MLPs do not support LoD representation and progressive transmission, which are critical to applications where adaptive resolution is essential, such as rendering acceleration or model compression.

To address this limitation, we investigate the relationship between the hidden representations within a single MLP and its final output. Our findings show that not only the last hidden representation but also earlier ones can produce effective signal representations when followed by an appropriate affine transformation. We also observe that, as depth increases, these hidden representations progressively capture higher-frequency components of the signal. This suggests that earlier hidden representations (i.e., those closer to the input) can serve as low-frequency approximations of the target signal.

Based on this observation, we propose the Tailed Multi-Layer Perceptron (T-MLP), a modified architecture of the classical MLP, to achieve LoD signal representation. Unlike the standard MLP that produces a single output only at the final layer, the T-MLP attaches an output branch, also called a *tail*, to each hidden layer. The first tail learns a coarse approximation of the target signal; the second tail captures the residual between the first output and the target; the third tail further refines the residual between the accumulated output and the target, and so on. That is, each tail is designed to focus on learning the residual between two consecutive levels of detail. Consequently, the T-MLP

naturally realizes LoD signal representation using supervision only from the highest-resolution signal.

Beyond LoD modeling, the T-MLP also supports progressive signal transmission: the parameters of the early layers, sufficient to generate the initial coarse output, can be transmitted first to a target device for rough rendering, while the parameters of subsequent layers are progressively delivered to gradually refine the signal representation according to the device's capability. We validate the effectiveness of T-MLP across a range of signal representation tasks and demonstrate its superiority over existing neural LoD baselines.

## 2 RELATED WORK

Our work is closely related to previous research on implicit neural representations and level of detail. In this section, we review some recent advances in these two areas.

**Implicit Neural Representations.** Representing shapes as continuous functions using Multi-Layer Perceptrons (MLPs) has attracted significant attention in recent years. Seminal methods encode shapes into latent codes, which are then concatenated with query coordinates and fed into a shared MLP to predict signed distances (Park et al., 2019; Chabra et al., 2020; Wang et al., 2023), occupancy values (Mescheder et al., 2019; Peng et al., 2020; Jiang et al., 2020), or unsigned distances (Chibane et al., 2020; Ren et al., 2023). Another line of work (Atzmon & Lipman, 2020; Gropp et al., 2020; Ma et al., 2020; Ben-Shabat et al., 2022; Yang et al., 2023; Zhou et al., 2024; Yang et al., 2025) focuses on overfitting a single 3D shape with carefully designed regularization terms to improve surface quality. Most of these methods adopt ReLU-based MLPs, which are known to suffer from a spectral bias toward low-frequency signals. To overcome this limitation, Fourier Features (Tancik et al., 2020) introduce a frequency-based encoding of inputs, while SIREN (Sitzmann et al., 2020) employs periodic activation functions and specialized initialization to better capture high-frequency details. MFN (Fathony et al., 2021) introduces a type of neural representation that replaces traditional layered depth with a multiplicative operation, but it lacks the inherent bias towards smoothness in both the represented function and its gradients. Other approaches explore combining explicit feature grids such as octrees (Takikawa et al., 2021; Yu et al., 2021) and hash tables (Müller et al., 2022) with MLPs to accelerate inference. However, these hybrid methods often incur significant memory overhead for high-fidelity geometry reconstruction. Beyond shape representation, implicit neural representations have been extended to encode images (Chen et al., 2021; Skorokhodov et al., 2021; Martel et al., 2021; He & Jin, 2024), videos (Sitzmann et al., 2020; Fathony et al., 2021; Yan et al., 2024), and textures (Oechsle et al., 2019; Henzler et al., 2020; Tu et al., 2024). Although these methods demonstrate impressive performance in signal representation, they are typically limited to capturing the signal at a single scale. In this work, we propose a novel architecture that learns multiple LoDs of the signal simultaneously and achieves superior performance compared to existing methods.

**Level of Detail.** Level of Detail (LoD) (Luebke et al., 2002) in computer graphics is widely used to reduce the complexity of 3D assets, aiming to improve efficiency in rendering or data transmission. Traditional geometry simplification methods (Hoppe, 1996; Garland & Heckbert, 1997; Szymczak et al., 2002; Surazhsky & Gotsman, 2003) focus on reducing polygon count by greedily removing mesh elements, while preserving the original mesh's geometric characteristics to the greatest extent possible. With the rise of INRs, several methods have explored LoD modeling in implicit representations. NGLOD (Takikawa et al., 2021) and MFLOD (Dou et al., 2023) leverage multilevel feature volumes to capture multiple LoDs, while PINs (Landgraf et al., 2022) introduce a progressive positional encoding scheme. BACON (Lindell et al., 2022) proposes band-limited coordinate-based networks to represent signals at multiple scales, but its performance is sensitive to the maximum bandwidth hyperparameter. ResidualMFN (Shekarforoush et al., 2022) introduces skip connections into MFN and proposes a novel initialization method for multi-scale signal representation. Mujkanovic et al. (2024) present Neural Gaussian Scale-Space Fields to learn continuous, anisotropic Gaussian scale spaces directly from raw data. Rebain et al. (2024) propose a novel formulation that unifies training and filtering as a maximum likelihood estimation problem, enabling neural fields to produce filtered versions of the training signal. BANF (Shabanov et al., 2024) adopts a cascaded training strategy to train multiple *independent* networks that progressively learn the residuals be-

tween the accumulated output and the ground-truth signal. In each stage of the cascade, BANF first queries a grid and then interpolates the grid values to obtain the output at the query point. To accurately represent the signal, very high-resolution grids are required, but querying such grids is extremely time-consuming and computationally expensive. In contrast, our method is designed based on the inherent properties of MLPs, enabling a *single* network to represent multiple LoDs with negligible computational overhead. It can seamlessly replace conventional MLPs in signal representation tasks.

## 3 OBSERVATIONS ABOUT MLP

The Multi-Layer Perceptron (MLP) is widely adopted in implicit neural representations (INRs), typically taking the following form:

$$
\begin{aligned}
\mathbf{h}_0 &= \mathbf{x}, \\
\mathbf{h}_i &= \sigma\left(\mathbf{W}_i \mathbf{h}_{i-1} + \mathbf{b}_i\right), i = 1, \ldots, k \\
\mathbf{y} &= \mathbf{W}^{out} \mathbf{h}_k + \mathbf{b}^{out},
\end{aligned}
\tag{1}
$$

where $\mathbf{x}$ is the input, $k$ denotes the number of hidden layers, $\mathbf{W}_i \in \mathbb{R}^{N_i \times M_i}$ and $\mathbf{b}_i \in \mathbb{R}^{N_i}$ define the affine transformation at the $i$-th hidden layer, and $\sigma$ denotes a nonlinear activation function. $\mathbf{W}^{out}$ and $\mathbf{b}^{out}$ represent the affine transformation in the output layer. In particular, the sinusoidal representation network (SIREN) (Sitzmann et al., 2020) employs the sine functions as the activation functions.

Although MLPs have demonstrated remarkable performance in INRs, they remain fundamentally limited in several aspects. First, MLPs output only a single representation at the last layer and thus do not inherently support multiple levels of detail (LoDs), which is a useful feature in data transmission and rendering for shape visualization. Second, a trained MLP for signal representation cannot be easily scaled in terms of its parameter size. In contrast, traditional mesh representations can utilize Progressive Mesh techniques (Hoppe, 1996) to construct a sequence of consecutive meshes from coarse to fine, which is crucial for controlling storage overhead and enabling progressive transmission. It should be noted that although many network compression techniques such as quantization (Yang et al., 2019; Lee et al., 2021; Xu et al., 2024) and pruning (Gao et al., 2021; Yeom et al., 2021; Gao et al., 2024) have been developed, they typically produce independent network copies. As a result, recording signal representations at multiple LoDs in this manner requires storing multiple networks simultaneously, leading to additional storage overhead.

To address this issue, we devised experiments to investigate the hidden representations at each layer within a single MLP. Our empirical findings indicate that, in addition to the final hidden representation, earlier hidden representations also provide meaningful approximations of the signal through an appropriate affine transformation. We also observe that these hidden representations tend to encode increasingly higher-frequency signal components as the network depth increases. Together, these findings suggest the possibility of using a single MLP to represent a signal at multiple LoDs. The experimental setup and corresponding results are detailed in Section 5.1.

As will be shown by our experiments, although the hidden representations at the early layers of an MLP tend to capture coarse-level information, the outputs derived from these hidden representations still fall significantly short of representing faithful low-detail signals. This is likely due to the lack of direct supervision, since the hidden layers are optimized only via backpropagation of gradients from the last output layer. In the next section, we will discuss how to address these limitations of MLP with a modified network structure and a new training strategy.

## 4 METHOD

### 4.1 TAILED MULTI-LAYER PERCEPTRON

To provide LoD signal representation, we propose the Tailed Multi-Layer Perceptron (T-MLP), as illustrated in Fig. 1. In contrast to standard MLPs that have a single output at the final layer, T-MLP attaches an output branch, also called a *tail*, to each hidden layer. Here, the output branch of the first layer is designed to learn a coarse approximation of the target signal, and the output branch of

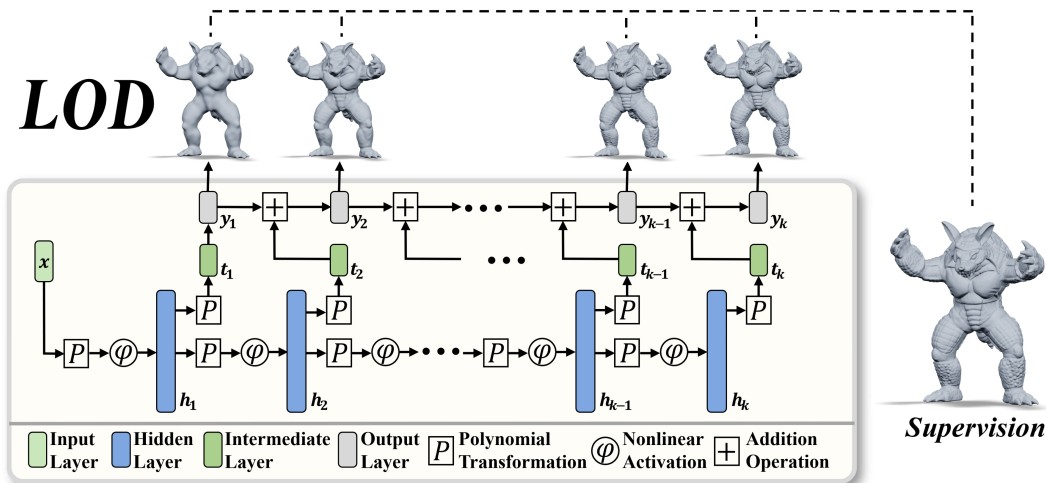

Figure 1: Overview of the T-MLP architecture. Built on a standard MLP, the T-MLP attaches an output branch, also called a *tail*, after each hidden layer. The first tail produces a coarse approximation of the target signal. The second tail learns the residual between the target and the first tail's output. The third tail captures the residual between the target and the cumulative output of the first two tails. In general, the $k$-th tail models the residual between the target signal and the sum of the outputs from the first $k-1$ tails.

each subsequent layer learns the residual between the output accumulated up to the previous layer and the ground truth supervision signal.

Formally, the architecture of the T-MLP is defined as:

$$
\begin{aligned}
\mathbf{h}_0 &= \mathbf{x}, \ \mathbf{h}_i = \sigma\left(\mathbf{W}_i \mathbf{h}_{i-1} + \mathbf{b}_i\right), \\
\mathbf{t}_i &= \mathbf{W}_i^{out} \mathbf{h}_i + \mathbf{b}_i^{out}, \\
\mathbf{y}_0 &= \mathbf{0}, \ \mathbf{y}_i = \mathbf{y}_{i-1} + \mathbf{t}_i, i = 1, \ldots, k.
\end{aligned}
\tag{2}
$$

Here, $\mathbf{t}_i$ denotes the intermediate output, i.e. residual prediction, at the $i$-th layer, and $\mathbf{y}_i$ represents the accumulated output up to that layer. Each output $\mathbf{y}_i$ is recursively obtained by adding the current intermediate prediction $\mathbf{t}_i$ to the previous output $\mathbf{y}_{i-1}$. This cumulative design enables each $\mathbf{t}_i$ for $i > 1$ to focus on learning the high-frequency components not yet captured, thereby preventing redundant learning of information already accounted for by previous outputs.

Because the magnitude of the residual is typically smaller than 1, the network would struggle to train properly with such significantly small magnitudes (Wang & Lai, 2024). Based on the simple fact that a value of a small magnitude can be expressed as the product of two values of larger magnitudes, we adopt a multiplicative formulation for $\mathbf{t}_i$ when $i > 1$ to mitigate this issue. Specifically, we set

$$
\begin{aligned}
\mathbf{t}_{i_0} &= \mathbf{W}_{i_0}^{out} \mathbf{h}_i + \mathbf{b}_{i_0}^{out}, \\
\mathbf{t}_{i_1} &= \mathbf{W}_{i_1}^{out} \mathbf{h}_i + \mathbf{b}_{i_1}^{out}, \\
\mathbf{t}_i &= \mathbf{t}_{i_0} \circ \mathbf{t}_{i_1}, i = 2, \ldots, k,
\end{aligned}
\tag{3}
$$

where $\circ$ stands for the Hadamard product, i.e., component-wise product. This multiplicative design can be interpreted as a low-rank quadratic transformation of the hidden representation $\mathbf{h}_i$ to produce the output $\mathbf{t}_i$, thereby enhancing the expressiveness of each output tail and improving the network's ability to fit residuals that are challenging for purely linear output layers. A detailed proof is provided in Appendix A.1.1.

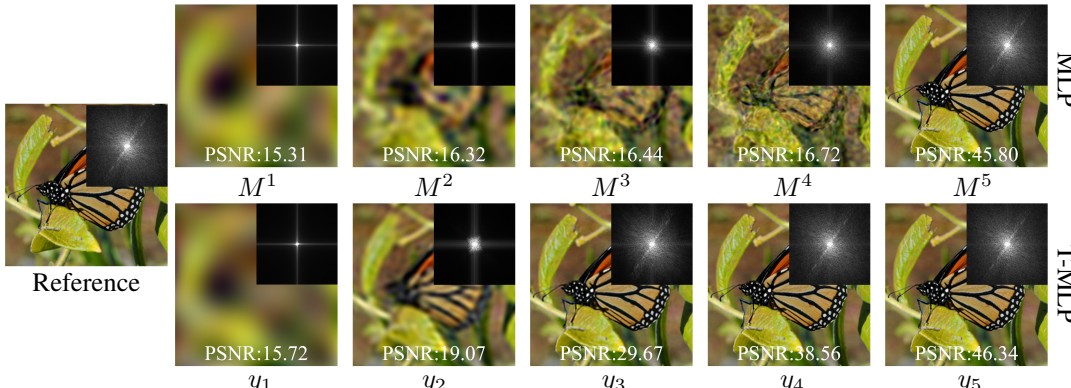

Figure 2: MLP VS T-MLP. The image is from the DIV2K dataset (Agustsson & Timofte, 2017) and has a resolution of $256 \times 256$.

## 4.2 TRAINING STRATEGY

We denote the original loss used to train a standard MLP as $\mathcal{L}$. For our proposed T-MLP, the training objective is defined as

$$\mathcal{L}_{total} = \sum_{i=1}^{k} \lambda_i \mathcal{L}(\mathbf{y}_i), \tag{4}$$

where $\mathbf{y}_i$ denotes the cumulative output up to the $i$-th output tail and $\lambda_i$ is a weighting coefficient that balances the losses from different output tails. Note that all tails are trained to approximate the same high-resolution target signal, without requiring any explicit supervision at multiple LoDs. This supervision strategy enables LoD representation because earlier tails, despite being supervised with high-resolution signals, possess limited parameter capacity and therefore can only reconstruct low-frequency components. As the network deepens, its representational capacity increases, allowing for the progressive refinement of high-frequency details.

Overall, our residual learning scheme enables the model to progressively approximate the target signal from coarse to fine, naturally supporting multiple LoDs. The multi-output design also allows the network to produce meaningful intermediate results without traversing the entire architecture, thereby enabling progressive transmission. Note that although both T-MLP and ResNet (He et al., 2016) leverage the concept of residuals, their underlying mechanisms differ fundamentally. A detailed comparison is provided in Section A.5.1 of the Appendix.

## 5 EXPERIMENTS

### 5.1 MLP VS T-MLP

To investigate the hidden representation at each layer within a single standard MLP, we design an experiment with the following procedure:

1. **Train the full model**: Train a standard MLP with $K$ hidden layers, denoted as $M^K$.

2. **Construct $M^{K-1}$**: Remove the final hidden and output layer of $M^K$, and attach a new linear output layer after the $(K-1)$-th hidden layer, resulting in an MLP with $K-1$ hidden layers, denoted as $M^{K-1}$.

3. **Train the new output layer**: Freeze the hidden layers of $M^{K-1}$ and retrain only the new-added linear output layer.

4. **Iterative procedure**: Repeat this process on $M^{K-1}$ to obtain $M^{K-2}$, and continue iteratively until $M^1$ is reached.

The first row of Fig. 2 shows the results of this procedure with $K = 5$ on an image fitting task using SIREN (Sitzmann et al., 2020). The results reveal that beyond the final hidden representation, earlier hidden representations can also approximate the signal through suitable affine transformations

and these hidden representations progressively capture higher frequency components as the network depth increases. These outputs from earlier-layer hidden representations can be viewed as low-detail approximations of the target signal, demonstrating the potential of a single MLP to represent multiple levels of detail (LoDs). However, there remains a significant gap between these intermediate outputs and satisfactory low-detail representations that could be expected.

The second row of Fig. 2 presents the outputs from each hidden representation of our proposed T-MLP. By attaching an output tail to every hidden layer, T-MLP enforces direct supervision at all layers to substantially improve the quality of intermediate representations. The layer-wise output branches of the T-MLP facilitate multiple LoDs and progressive transmission.

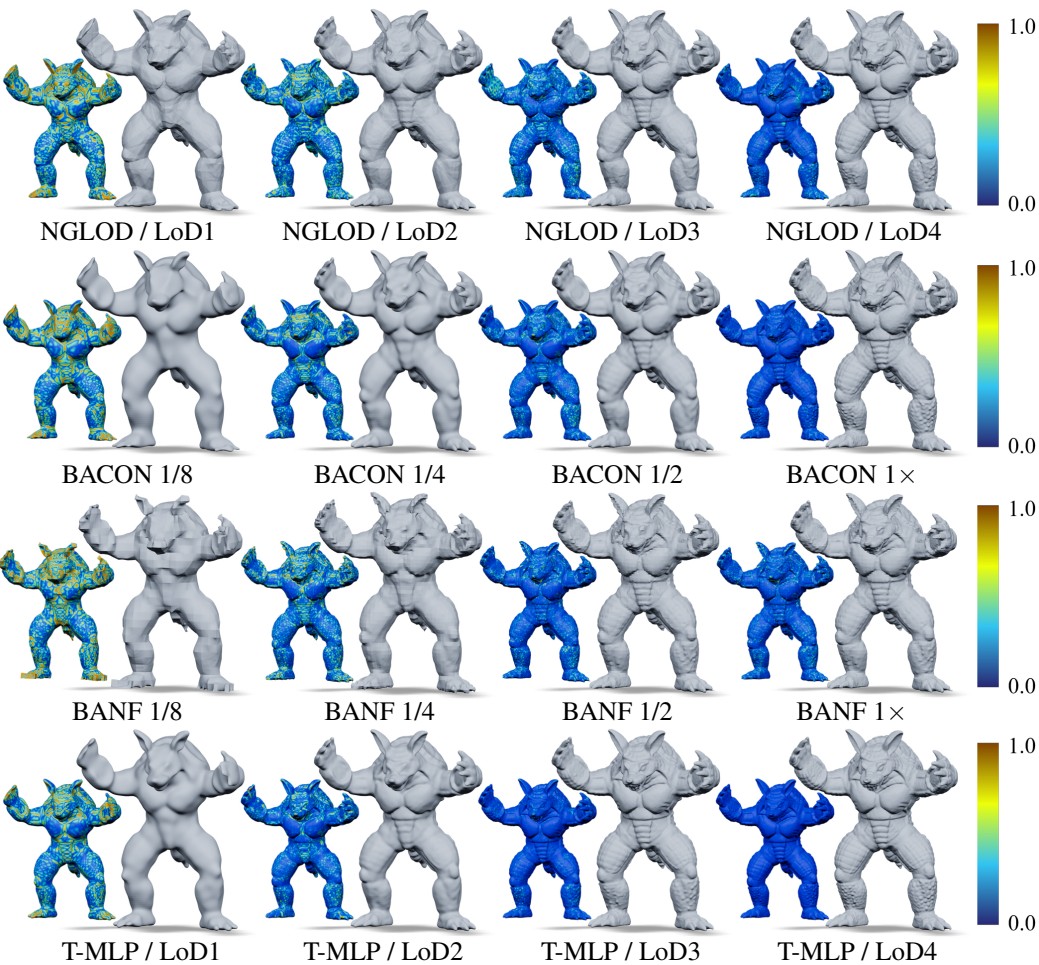

Figure 3: Visual comparisons between our T-MLP and the baseline methods for 3D shape LoD representation. (Additional comparisons are provided in Section A.2.3 of the Appendix.)

## 5.2 LoD Signal Representation

To evaluate the effectiveness of T-MLP, we compare it on both 3D shape representation and image representation tasks with several baseline methods: Fourier Features (Tancik et al., 2020), SIREN (Sitzmann et al., 2020), NGLOD (Takikawa et al., 2021), BACON (Lindell et al., 2022), and BANF (Shabanov et al., 2024). Among them, Fourier Features and SIREN do not support LoD, while NGLOD, BACON, and BANF are designed with LoD mechanisms. Since BANF has not released its code for the 3D shape representation task, we reimplemented it based on the paper for this task. Results of the other baseline methods are obtained from their official open-source implementations.

Table 1: Quantitative comparisons for 3D shape representation across multiple LoDs. Methods not appearing in lower LoDs do not support LoD.

| LoD | Method | Thingi10K | | | | Stanford 3D Scanning Repository | | | |
|---|---|---|---|---|---|---|---|---|---|
| | | CD ↓ | | NC ↑ | | CD ↓ | | NC ↑ | |
| | | Mean | Median | Mean | Median | Mean | Median | Mean | Median |
| LoD4 | Fourier Features (Tancik et al., 2020) | 1.871 | 1.866 | 98.22 | 98.39 | 1.763 | 1.783 | 95.52 | 97.29 |
| | SIREN (Sitzmann et al., 2020) | 1.769 | 1.763 | 99.19 | 99.23 | 1.613 | 1.611 | 96.90 | 98.73 |
| | NGLOD (Takikawa et al., 2021) | 1.975 | 1.877 | 99.02 | 99.22 | 1.711 | 1.736 | 96.86 | 98.52 |
| | BACON (Lindell et al., 2022) | 1.787 | 1.777 | 99.06 | 99.13 | 1.638 | 1.666 | 96.63 | 98.55 |
| | BANF (Shabanov et al., 2024) | 4.683 | 3.191 | 96.08 | 96.81 | 1.870 | 1.859 | 94.82 | 96.73 |
| | Ours | **1.740** | **1.731** | **99.39** | **99.44** | **1.513** | **1.460** | **98.03** | **99.11** |
| LoD3 | NGLOD (Takikawa et al., 2021) | 2.148 | 2.034 | 98.55 | 98.77 | 2.078 | 2.100 | 94.89 | 97.14 |
| | BACON (Lindell et al., 2022) | 1.999 | 1.962 | 98.18 | 98.50 | 2.145 | 2.194 | 93.75 | 93.85 |
| | BANF (Shabanov et al., 2024) | 4.437 | 3.153 | 96.18 | 97.09 | 1.906 | 1.874 | 94.24 | 96.02 |
| | Ours | **1.771** | **1.761** | **99.20** | **99.25** | **1.615** | **1.638** | **97.01** | **98.77** |
| LoD2 | NGLOD (Takikawa et al., 2021) | 2.587 | 2.384 | 97.54 | 97.52 | 2.821 | 2.836 | 92.12 | 94.37 |
| | BACON (Lindell et al., 2022) | 2.200 | 2.096 | 97.51 | 97.94 | 2.607 | 2.452 | 91.68 | 93.73 |
| | BANF (Shabanov et al., 2024) | 6.660 | 5.183 | 93.69 | 94.82 | 2.785 | 2.804 | 89.72 | 90.96 |
| | Ours | **1.949** | **1.926** | **98.45** | **98.53** | **2.042** | **2.072** | **94.36** | **96.53** |
| LoD1 | NGLOD (Takikawa et al., 2021) | 3.545 | 3.385 | 95.62 | 96.24 | 4.246 | 4.265 | 87.91 | 89.35 |
| | BACON (Lindell et al., 2022) | 3.041 | 2.907 | 95.56 | 96.20 | 4.451 | 4.203 | 85.98 | 85.82 |
| | BANF (Shabanov et al., 2024) | 8.611 | 7.234 | 90.76 | 91.63 | 5.061 | 5.314 | 83.19 | 83.83 |
| | Ours | **2.587** | **2.443** | **96.56** | **97.28** | **3.423** | **3.220** | **89.07** | **90.53** |

### 5.2.1 3D SHAPE REPRESENTATION

We use 3D models from the Thingi32 subset of Thingi10K (Zhou & Jacobson, 2016) and the Stanford 3D Scanning Repository to learn Signed Distance Functions (SDFs) at multiple levels of detail (LoDs). T-MLP, configured with five hidden layers of 256 units each, is employed to fit the SDF. It adopts sine activation and follows the initialization strategy proposed in SIREN (Sitzmann et al., 2020). Following the baseline settings, we set the number of LoDs to 4, with output tail weights defined as $(\lambda_1, \lambda_2, \lambda_3, \lambda_4, \lambda_5) = (0, 0.5, 0.5, 0.5, 2.5)$. The loss is formulated as:

$$\mathcal{L}_{sdf} = \sum_{i=1}^{5} \frac{\lambda_i}{|\mathcal{Q}|} \sum_{\mathbf{x} \in \mathcal{Q}} |y_i(\mathbf{x}) - y_{gt}(\mathbf{x})|, \tag{5}$$

where $y_i$ denotes the cumulative output up to the $i$-th output tail, $y_{gt}$ denotes the ground-truth SDF value, and $\mathcal{Q}$ represents the set of sampled query points. We extract meshes from the SDFs using the Marching Cubes algorithm (Lorensen & Cline, 1987) with a grid resolution of $512^3$. For evaluation, we uniformly sample 500k points from each mesh and compute the Chamfer Distance (CD) and Normal Consistency (NC). Please refer to Section A.2.1 of Appendix for additional implementation details.

We provide quantitative and qualitative comparisons in Tab. 1 and Fig. 3, with additional results in Section A.2.3 of the Appendix. NGLOD requires a large number of parameters to achieve satisfactory shape representation. For BACON, we observe that its performance is highly sensitive to the maximum bandwidth hyperparameter: a small value leads to overly smooth shapes, while a large value results in rough and irregular geometry. BANF incurs high computational costs due to querying multiple $N^3$ grids at different resolutions and struggles to capture shape features, especially on the Thingi10K dataset; please refer to the Appendix for visual results. In addition, BANF employs a separate network at each stage to incrementally learn residuals with respect to the target signal, which leads to increased parameter count and longer training times.

In contrast, our method builds upon the inherent properties of MLPs and introduces architectural modifications that enable a single network to represent and train multiple LoDs simultaneously. T-MLP consistently achieves higher representation accuracy across all LoDs. We also observe that T-MLP surpasses standard MLP (i.e., SIREN) at the highest LoD, which we attribute to its ability to supervise all hidden layers, leading to more stable and effective optimization, rather than relying solely on backpropagation to indirectly adjust the parameters of earlier layers.

Additionally, we can obtain *continuous* LoDs by interpolating between discrete LoDs. Please refer to Section A.2.2 of the Appendix for details. We report the parameter count and training time of

each method in Tab. 2. While our method is slower than those that do not support LoD, it is faster than the methods that support LoD, particularly NGLOD and BANF by a large margin.

Table 2: Runtime (in minutes) and parameter counts for learning a single shape.

|  | Fourier Features | SIREN | NGLOD | BACON | BANF | T-MLP (Ours) |
|---|---|---|---|---|---|---|
| LoD | ✗ | ✗ | ✓ | ✓ | ✓ | ✓ |
| #Params | 263k | 265k | 1.35M | 264k | 2.08M | 266k |
| Time (min) | 0.815 | 2.988 | 44.80 | 6.217 | 67.31 | 3.548 |

Implicit neural representations are also widely used to reconstruct continuous surfaces from point clouds. In Section A.2.4 of the Appendix, we present the results of our T-MLP on surface reconstruction from point clouds, demonstrating that our low-LoD outputs effectively resist noise through underfitting on noisy point clouds, while high-LoD representations can accurately recover fine geometric details when the data is clean.

### 5.2.2 IMAGE REPRESENTATION

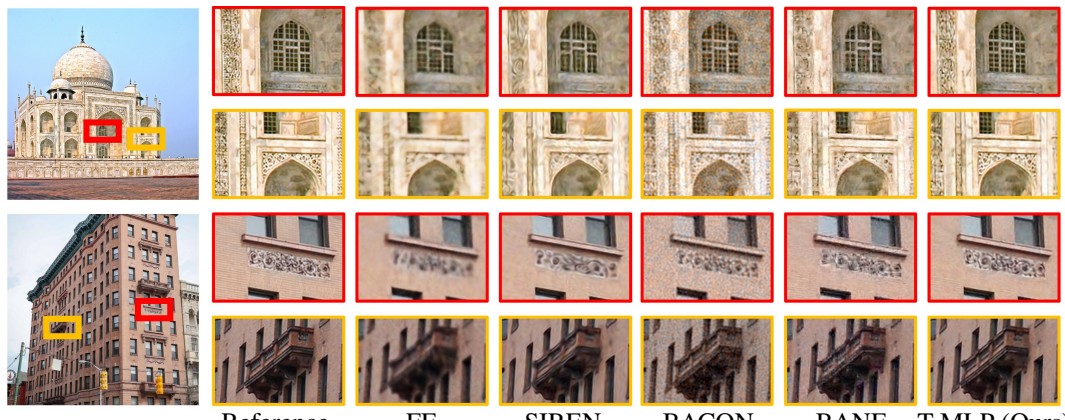

Reference      FF      SIREN      BACON      BANF      T-MLP (Ours)

Figure 4: Visual comparisons of image fitting at the highest LoD with a resolution of $1024 \times 1024$.

Table 3: Quantitative results for image fitting across multiple LoDs on the DIV2K dataset. Methods not appearing in lower LoDs do not support LoD.

|  | Method | $512 \times 512$ | | | | $1024 \times 1024$ | | | |
|---|---|---|---|---|---|---|---|---|---|
|  |  | PSNR ↑ | | SSIM ↑ | | PSNR ↑ | | SSIM ↑ | |
|  |  | Mean | Median | Mean | Median | Mean | Median | Mean | Median |
| | Fourier Features | 29.39 | 28.72 | 90.09 | 89.49 | 25.81 | 25.46 | 77.73 | 77.70 |
| | SIREN | 33.39 | 33.88 | 94.18 | 93.82 | 28.02 | 27.83 | 83.83 | 84.67 |
| LoD3 | BACON (Lindell et al., 2022) | 31.73 | 31.55 | 89.81 | 90.18 | 24.43 | 24.00 | 58.20 | 57.65 |
| | BANF (Shabanov et al., 2024) | 32.46 | 32.07 | 95.40 | 95.29 | 27.39 | 27.42 | 85.48 | 86.35 |
| | T-MLP | **35.92** | **36.07** | 95.31 | **95.67** | **30.22** | **29.64** | **86.22** | **86.65** |
| | BACON (Lindell et al., 2022) | 25.93 | 25.70 | 79.04 | 78.82 | 21.76 | 21.55 | 47.19 | 46.64 |
| LoD2 | BANF (Shabanov et al., 2024) | 25.61 | 25.33 | 82.72 | 81.96 | 24.25 | 24.16 | 72.89 | 72.80 |
| | T-MLP | **31.49** | **31.85** | **91.47** | **91.71** | **26.42** | **26.61** | **77.63** | **78.34** |
| | BACON (Lindell et al., 2022) | 23.08 | 22.62 | 65.37 | 64.20 | 20.79 | 20.43 | 42.55 | 43.58 |
| LoD1 | BANF (Shabanov et al., 2024) | 22.75 | 22.30 | 67.77 | 66.45 | **22.30** | 22.06 | **61.10** | **61.50** |
| | T-MLP | **23.69** | **23.59** | **69.01** | **68.47** | 22.04 | **22.10** | 57.45 | 56.34 |

We also evaluate the performance of T-MLP on the image fitting task. We select images from the DIV2K dataset (Agustsson & Timofte, 2017) with resolutions of $512 \times 512$ and $1024 \times 1024$ for both quantitative and qualitative comparisons. T-MLP is trained with five hidden layers of 256 units each using the Adam optimizer for 10k iterations. Consistent with the baseline settings, the number of LoDs is set to 3, and the output tail weights are set as $(\lambda_1, \lambda_2, \lambda_3, \lambda_4, \lambda_5) = (0, 0, 1, 1, 1)$. The

loss is formulated as:

$$\mathcal{L}_{image} = \sum_{i=1}^{5} \frac{\lambda_i}{N} \sum_{\mathbf{x}} \|\mathbf{y}_i(\mathbf{x}) - \mathbf{y}_{gt}(\mathbf{x})\|_2^2 , \qquad (6)$$

where $\mathbf{y}_i$ represents the $i$-th output of the network, $\mathbf{y}_{gt}$ denotes the ground-truth RGB color, and $N$ represents the number of pixels.

The visual comparisons in Fig. 4 and the quantitative results in Tab. 3 demonstrate that T-MLP achieves more accurate image representation at both resolutions ($512^2$ and $1024^2$) across different LoDs. Additionally, we present image fitting results on images corrupted with Gaussian noise in Section A.3.3 of the Appendix, showing that our low-detail representations effectively suppress high-frequency noise components.

To further evaluate the generality of our method, we also conduct experiments on neural radiance field representation and present the results in Section A.4 of the Appendix.

## 5.3 Ablation Studies

**Effect of the Residual Design.**   To evaluate the effectiveness of the residual design in T-MLP, we make each output tail directly learn the ground-truth signal rather than learning the residual, and conduct experiments on 3D shape representation using the Stanford 3D Scanning Repository. The quantitative comparisons in Tab. 4 show that T-MLP without the residual design is less effective than our version with it. This is because the residual formulation enables the later hidden representations to focus on learning the residuals between the current approximation and the ground-truth signal, avoiding redundantly learning the information already encoded by earlier layers.

In Section A.5.1 of the Appendix, we also present a comparison with MLPs with residual connections (He et al., 2016) to show the differences and advantages of our approach over ResNet.

**Effect of the Multiplicative Design.**   We conduct experiments to verify the effectiveness of the multiplicative design in Eq. 3. As illustrated in Tab. 4, incorporating the multiplicative design leads to more accurate 3D shape representations compared to the baseline without it.

Table 4: Effect of the Residual Design and Multiplicative Design.

| Network | CD $\downarrow$ | NC $\uparrow$ |
|---|---|---|
| T-MLP w/o Residual Design | 1.582 | 97.52 |
| T-MLP w/o Multiplicative Design | 1.521 | 97.94 |
| Full T-MLP (Ours) | **1.513** | **98.03** |

## 6 Discussion and Conclusion

In this paper, we have found that, within a single MLP, not only the final hidden representation but also earlier hidden representations provide meaningful approximations of the signal through appropriate affine transformations, and that these representations tend to encode progressively higher-frequency components as network depth increases. Based on this observation, we have proposed the Tailed Multi-Layer Perceptron (T-MLP), an enhanced MLP architecture that attaches an output tail to each hidden layer. Each tail incrementally learns the residual between the current approximation and the ground-truth signal, enabling the network to support multiple levels of detail (LoDs) and progressive transmission. Across various signal representation tasks, T-MLP demonstrates superior performance compared to existing neural LoD baselines.

**Limitations and Future Work.**   Although T-MLP enables LoD representation, it remains unclear how deep or wide a network is required to accurately represent a given signal. For instance, in an $N$-layer T-MLP, if the first $M$ layers ($M < N$) already capture the signal sufficiently, the subsequent layers may only preserve the existing performance without learning additional high-frequency details, leading to redundant parameters. One promising direction is to integrate pruning into training by monitoring whether a layer has already fully represented the target signal; once this condition is met, the subsequent layers can be removed to avoid parameter redundancy.

**Reproducibility Statement.** We are committed to ensuring the reproducibility of our findings. The proposed method is described in detail in Section 4, while the network architecture, loss functions, hyperparameter settings, and other experimental configurations are provided in Section 5 and Appendix A.2.1. All datasets used in our experiments are publicly available and properly cited. The source code will be released upon acceptance.

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

# A APPENDIX

## A.1 TAILED MULTI-LAYER PERCEPTRON

### A.1.1 MULTIPLICATIVE DESIGN

The multiplicative design defined in Eq. 3 of the main paper is given as:

$$
\begin{aligned}
\mathbf{t}_{i_0} &= \mathbf{W}_{i_0}^{out}\mathbf{h}_i + \mathbf{b}_{i_0}^{out}, \\
\mathbf{t}_{i_1} &= \mathbf{W}_{i_1}^{out}\mathbf{h}_i + \mathbf{b}_{i_1}^{out}, \\
\mathbf{t}_i &= \mathbf{t}_{i_0} \circ \mathbf{t}_{i_1}, i = 2, \ldots, k,
\end{aligned}
\tag{1}
$$

where $\mathbf{W}_{i_0}^{out} \in \mathbb{R}^{D \times N_i}$, $\mathbf{W}_{i_1}^{out} \in \mathbb{R}^{D \times N_i}$, $\mathbf{b}_{i_0}^{out} \in \mathbb{R}^D$ and $\mathbf{b}_{i_1}^{out} \in \mathbb{R}^D$. Here, $D$ is the dimension of output $\mathbf{t}_i$ and $N_i$ denotes the dimension of the $i$-th hidden representation $\mathbf{h}_i$. For clarity, consider the case where the output $t_i$ is a scalar. Let $\mathbf{a}^\top = W_{i_0}^{out} \in \mathbb{R}^{1 \times N_i}$, $\mathbf{b}^\top = W_{i_1}^{out} \in \mathbb{R}^{1 \times N_i}$, $\mathbf{x} = \mathbf{h}_i \in \mathbb{R}^{N_i \times 1}$, $c = \mathbf{b}_{i_0}^{out} \in \mathbb{R}$ and $d = \mathbf{b}_{i_1}^{out} \in \mathbb{R}$. Then the output $t_i$ can be rewritten as:

$$
t_i = (\mathbf{a}^\top\mathbf{x} + c)(\mathbf{b}^\top\mathbf{x} + d) = (\mathbf{a}^\top\mathbf{x})(\mathbf{b}^\top\mathbf{x}) + d(\mathbf{a}^\top\mathbf{x}) + c(\mathbf{b}^\top\mathbf{x}) + cd.
\tag{2}
$$

Alternatively, this expression can be written in compact matrix form as:

$$
t_i = \mathbf{x}^\top Q\mathbf{x} + \mathbf{u}^\top\mathbf{x} + s,
\tag{3}
$$

where $Q = \mathbf{a}\mathbf{b}^\top \in \mathbb{R}^{N_i \times N_i}$, $\mathbf{u}^\top = d\mathbf{a}^\top + c\mathbf{b}^\top \in \mathbb{R}^{1 \times N_i}$, and $s = cd \in \mathbb{R}$.

This formulation shows that T-MLP implements a low-rank quadratic transformation of the hidden representation $\mathbf{x}$ (i.e., $\mathbf{h}_i$) to produce the output $t_i$. In the case where $t_i$ is multi-dimensional, the same operation is applied independently to each output dimension.

## A.2 3D SHAPE REPRESENTATION

### A.2.1 IMPLEMENTATION DETAILS

We use T-MLP with five hidden layers, each containing 256 hidden features, to fit SDF. T-MLP adopts the sine activation function and follows the initialization strategy proposed in SIREN (Sitzmann et al., 2020). The Adam optimizer is used with the initial learning rate of $3 \times 10^{-4}$ and training is run for 10k iterations. The learning rate decays by a factor of 0.25 at the 7000th, 8000th, and 9000th iterations.

All shapes are normalized to fit within the bounding box $[-1, 1]^3$. During each training iteration, we sample 100k training points: 20% are randomly sampled from the bounding box, 40% are surface points, and the remaining 40% are near-surface points, obtained by perturbing the surface points with Gaussian noise ($\sigma = 0.01$). The loss is formulated as:

$$
\mathcal{L}_{sdf} = \sum_{i=1}^{5} \frac{\lambda_i}{|\mathcal{Q}|} \sum_{\mathbf{x} \in \mathcal{Q}} |y_i(\mathbf{x}) - y_{gt}(\mathbf{x})|,
\tag{4}
$$

where $y_i$ represents the cumulative output up to the $i$-th output tail, $y_{gt}$ denotes the ground-truth SDF value, and $\mathcal{Q}$ represents the set of sampled query points. The output tail weights are set as $(\lambda_1, \lambda_2, \lambda_3, \lambda_4, \lambda_5) = (0, 0.5, 0.5, 0.5, 2.5)$.

Meshes are extracted from the predicted SDFs using the Marching Cubes algorithm (Lorensen & Cline, 1987) with a grid resolution of $512^3$. For evaluation, 500k points are uniformly sampled from each mesh, and Chamfer Distance (CD) and Normal Consistency (NC) are computed.

### A.2.2 CONTINUOUS LODs

We can generate a continuous 3D shape transition from the lowest to the highest level of detail (LoD) by interpolating between adjacent LoDs. Specifically, an arbitrary LoD $l$ is computed using the following interpolation formula:

$$
\begin{aligned}
y_l &= y_{l^*} + \alpha t_{l^*+1} \\
&= (1 - \alpha)y_{l^*} + \alpha y_{l^*+1}
\end{aligned}
\tag{5}
$$

where $l^* = \lfloor l \rfloor$ and $\alpha = l - \lfloor l \rfloor$. Fig. A1 shows the resulting continuous LoDs for the Happy Buddha model from the Stanford 3D Scanning Repository.

### A.2.3 ADDITIONAL RESULTS

We provide additional visual results of 3D shape representation in Figs. A2, A3, and A4. Experimental results demonstrate that our method consistently outperforms all baselines across different LoDs. BANF (Shabanov et al., 2024) struggles to model shape features, resulting in poor performance on the Thingi10K dataset (Zhou & Jacobson, 2016). In some cases, its outputs at higher LoDs even underperform compared to those at lower LoDs.

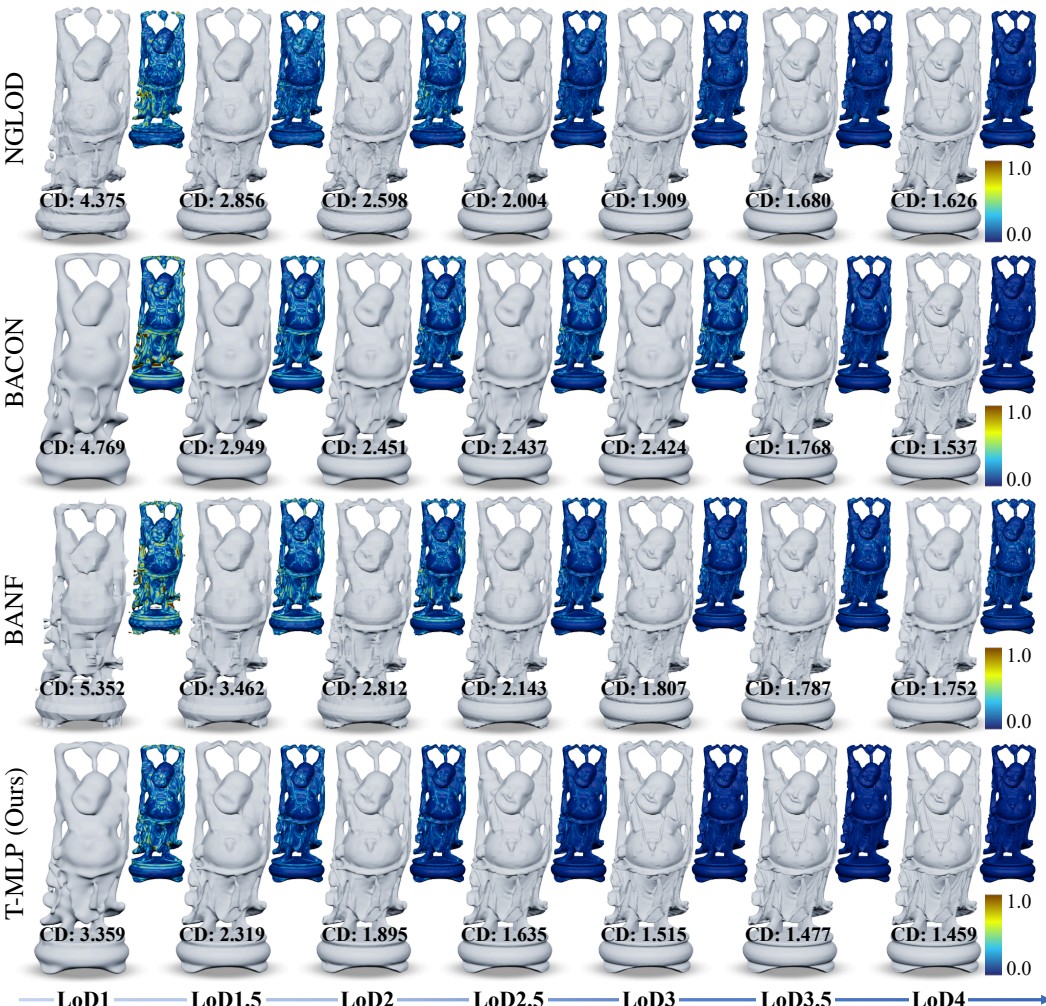

Figure A1: Visual comparisons between our T-MLP and the baseline methods for continuous LoDs. Zoom in to see details.

### A.2.4 SURFACE REONSTRUCTION FROM POINT CLOUD

When reconstructing continuous surfaces from point clouds, some methods attempt to fully fit the point cloud to recover fine geometric details. However, this often leads to overfitting in the presence of noise, resulting in overly jagged or unsatisfactory surfaces. Denoising techniques typically impose smoothness constraints but risk oversmoothing fine structures. Moreover, without access to the ground-truth surface, it is inherently ambiguous to determine whether a point cloud contains noise, as the target surface may itself be non-smooth.

Our T-MLP's LoD representation naturally addresses this challenge: high-detail outputs capture fine geometry in clean data, while lower-detail outputs suppress noise through underfitting. To validate this, we perform experiments on the Stanford 3D Scanning Repository using the loss function

from StEik (Yang et al., 2023) that introduces a second-order constraint to enhance stability and convergence when learning SDFs from unoriented point clouds. As shown in the first row of Fig. A5, T-MLP successfully reconstructs fine geometric details from clean point clouds. In the second row, results on noisy inputs demonstrate that its low-detail outputs effectively reduce noise while preserving the overall shape.

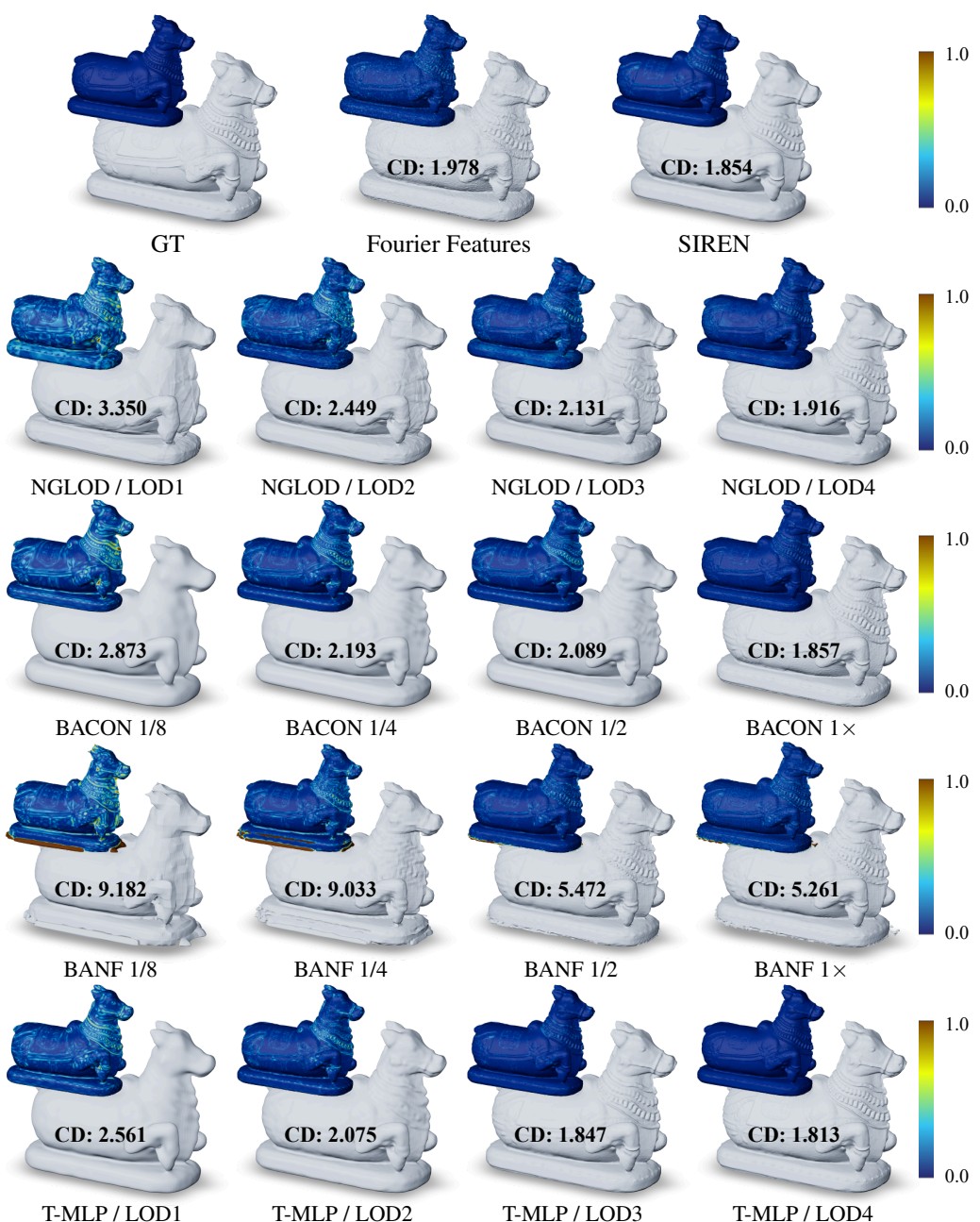

Figure A2: Visual comparisons between our T-MLP and the baseline methods for 3D shape LoD representation.

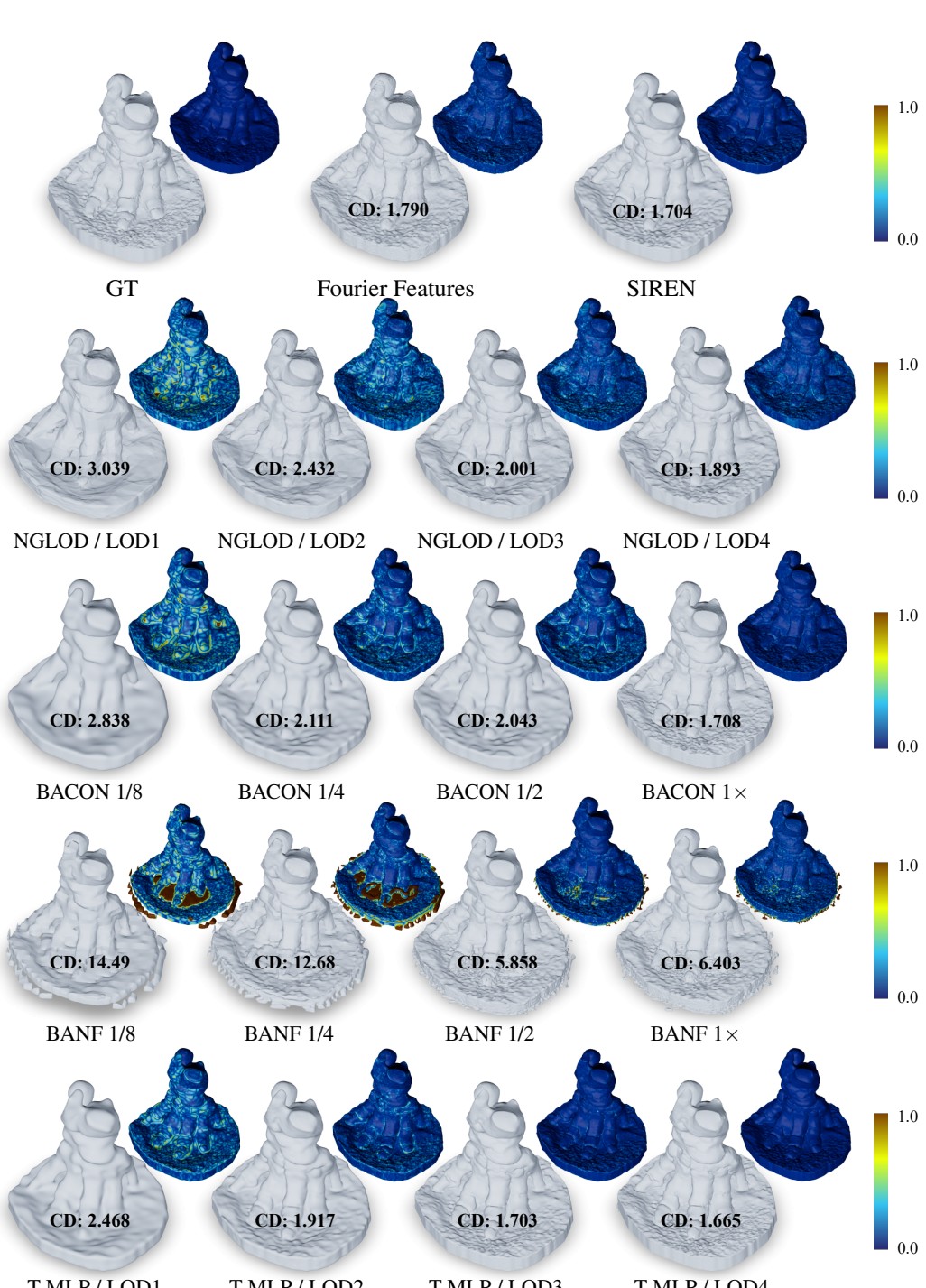

Figure A3: Visual comparisons between our T-MLP and the baseline methods for 3D shape LoD representation.

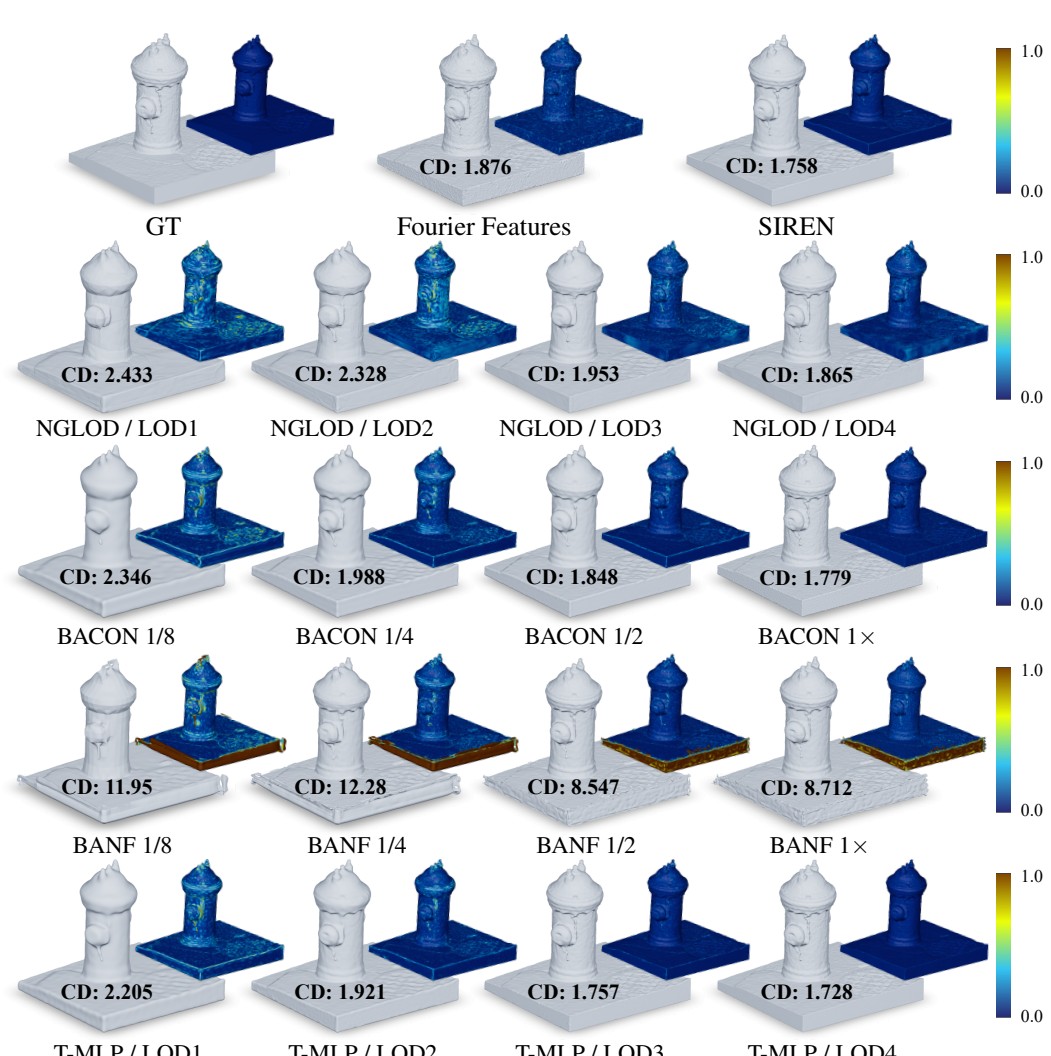

Figure A4: Visual comparisons between our T-MLP and the baseline methods for 3D shape LoD representation.

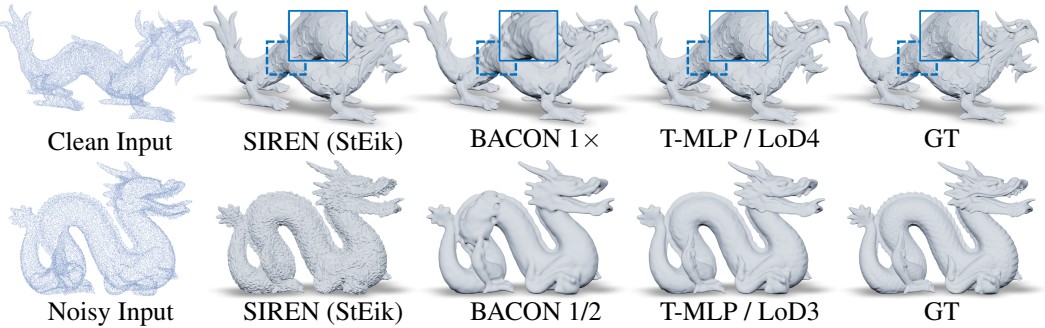

Figure A5: Visual comparisons between our T-MLP and the baseline methods for surface reconstruction from point clouds on the Stanford 3D Scanning Repository.

### A.3 IMAGE REPRESENTATION

#### A.3.1 IMPLEMENTATION DETAILS

#### A.3.2 ADDITIONAL RESULTS

We present visual comparisons in Fig. A7 on clean image representation task across multiple LoDs.

#### A.3.3 NOISY IMAGE FITTING

We add Gaussian noise with a standard deviation of 15 to images from the DIV2K dataset (Agustsson & Timofte, 2017), and use the resulting noisy images as supervision signals for training. The number of LoDs is set to 4. As shown in Fig. A6, the low-detail outputs of T-MLP effectively suppress high-frequency noise components through underfitting.

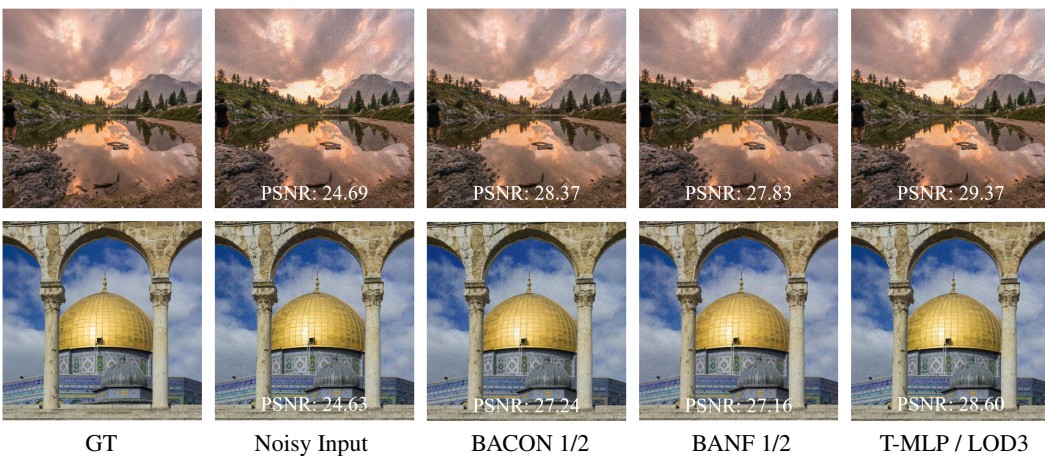

| GT | Noisy Input | BACON 1/2 | BANF 1/2 | T-MLP / LOD3 |

Figure A6: Visual comparisons of noisy image fitting. The resolution of the images is $512 \times 512$.

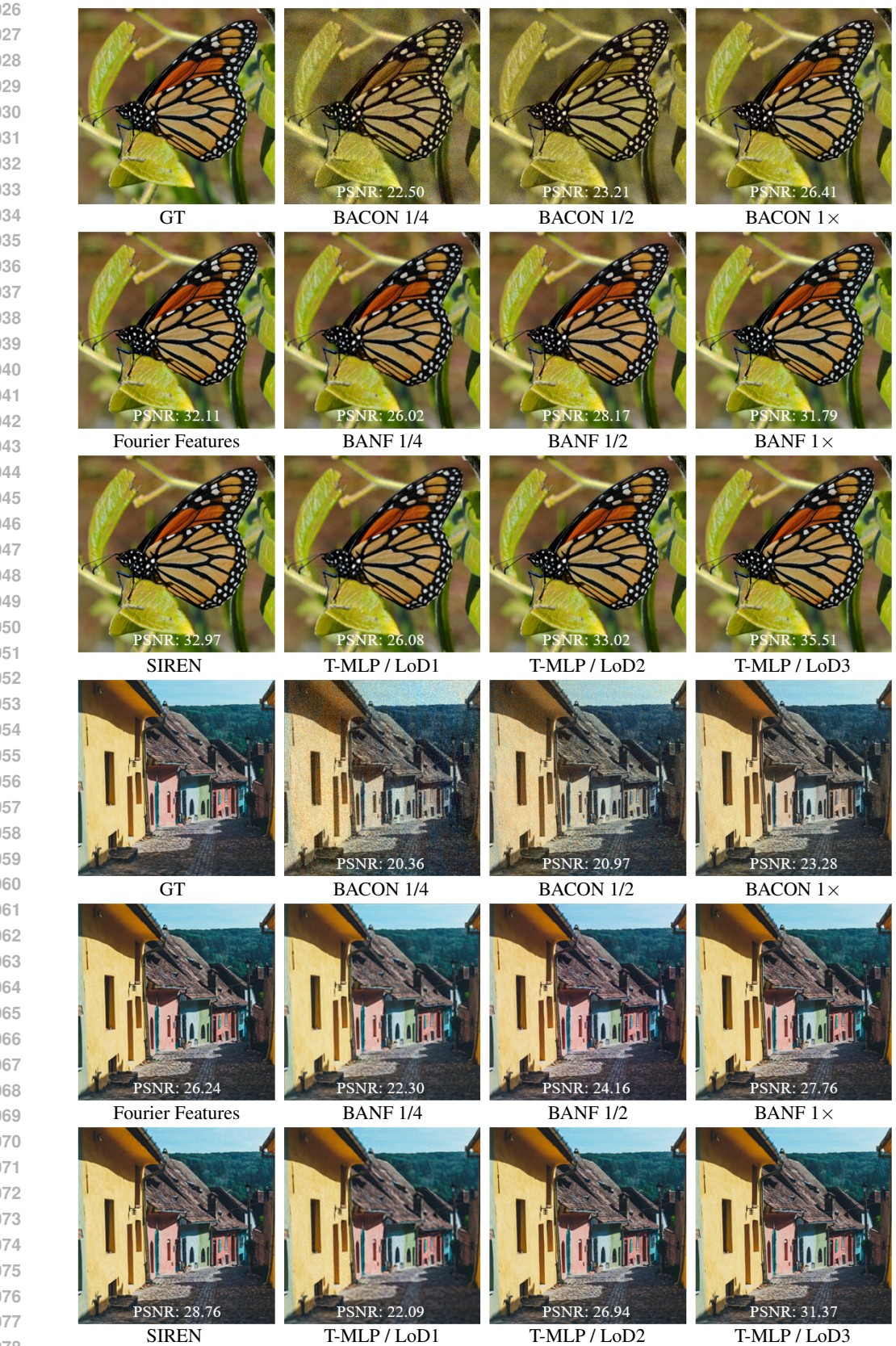

Figure A7: Visual comparisons of image fitting on the DIV2K dataset (Agustsson & Timofte, 2017) with a resolution of $1024 \times 1024$.

## A.4 NEURAL RADIANCE FIELD

Given a set of multi-view images with known camera poses, Neural Radiance Fields (NeRF) (Mildenhall et al., 2021) represent each image pixel as a ray:

$$\mathbf{r}(t) = \mathbf{o} + t\mathbf{d}, \tag{6}$$

where $\mathbf{o}$ is the camera origin and $\mathbf{d}$ is the direction vector passing through the pixel. To predict the pixel color $\mathbf{C}(\mathbf{r})$, NeRF uses the volume rendering equation by integrating predicted color $\mathbf{c}$ and density $\sigma$ along the ray. Specifically, a neural network is queried at sampled positions along the ray to obtain values $\mathbf{c}_j$ and $\sigma_j$, and the final color is computed as:

$$\mathbf{C}(\mathbf{r}) = \sum_j T_j \left(1 - \exp\left(-\sigma_j(t_{j+1} - t_j)\right)\right) \mathbf{c}_j, \tag{7}$$

$$T_j = \exp\left(-\sum_{i<j} \sigma_i(t_{i+1} - t_i)\right), \tag{8}$$

where $T_j$ denotes the accumulated transmittance up to sample $j$. The expression

$$w_j = T_j \left(1 - \exp\left(-\sigma_j(t_{j+1} - t_j)\right)\right) \tag{9}$$

can be interpreted as alpha compositing weights for the corresponding color $\mathbf{c}_j$.

To evaluate the effectiveness of T-MLP in neural radiance field fitting, we conduct experiments on the Blender dataset (Mildenhall et al., 2021), using BACON (Lindell et al., 2022) as the baseline. We use the Adam optimizer with an initial learning rate of $5 \times 10^{-4}$ to train T-MLP with 5 hidden layers and 256 hidden features per layer. Training is conducted for 10k iterations, with the learning rate decaying by a factor of 0.25 every 2k iterations. We also train BACON for 10k iterations to match our method. Visual results are shown in Figure A8. Experimental results demonstrate that T-MLP consistently outperforms BACON across all levels of detail (LoDs).

Following the supervision strategy in BACON (Lindell et al., 2022), we also evaluate T-MLP on the multiscale Blender dataset (Mildenhall et al., 2021), which contains images at multiple resolutions, including 512×512, 256×256, 128×128, and 64×64. In this setting, the four outputs $y_i$ of T-MLP ($i \in [1, 2, 3, 4]$) are supervised using ground-truth images at 1/8, 1/4, 1/2, and full resolution, respectively. Unlike the single-scale supervision used in the neural radiance field fitting task above, where all outputs are trained against the same ground-truth image, this task employs a multiscale supervision scheme, assigning different resolution targets to different outputs. As illustrated in Fig. A9, T-MLP consistently outperforms BACON under this multiscale setting. Note that the quantitative results in Fig. A9 are evaluated against ground-truth images at the corresponding resolutions.

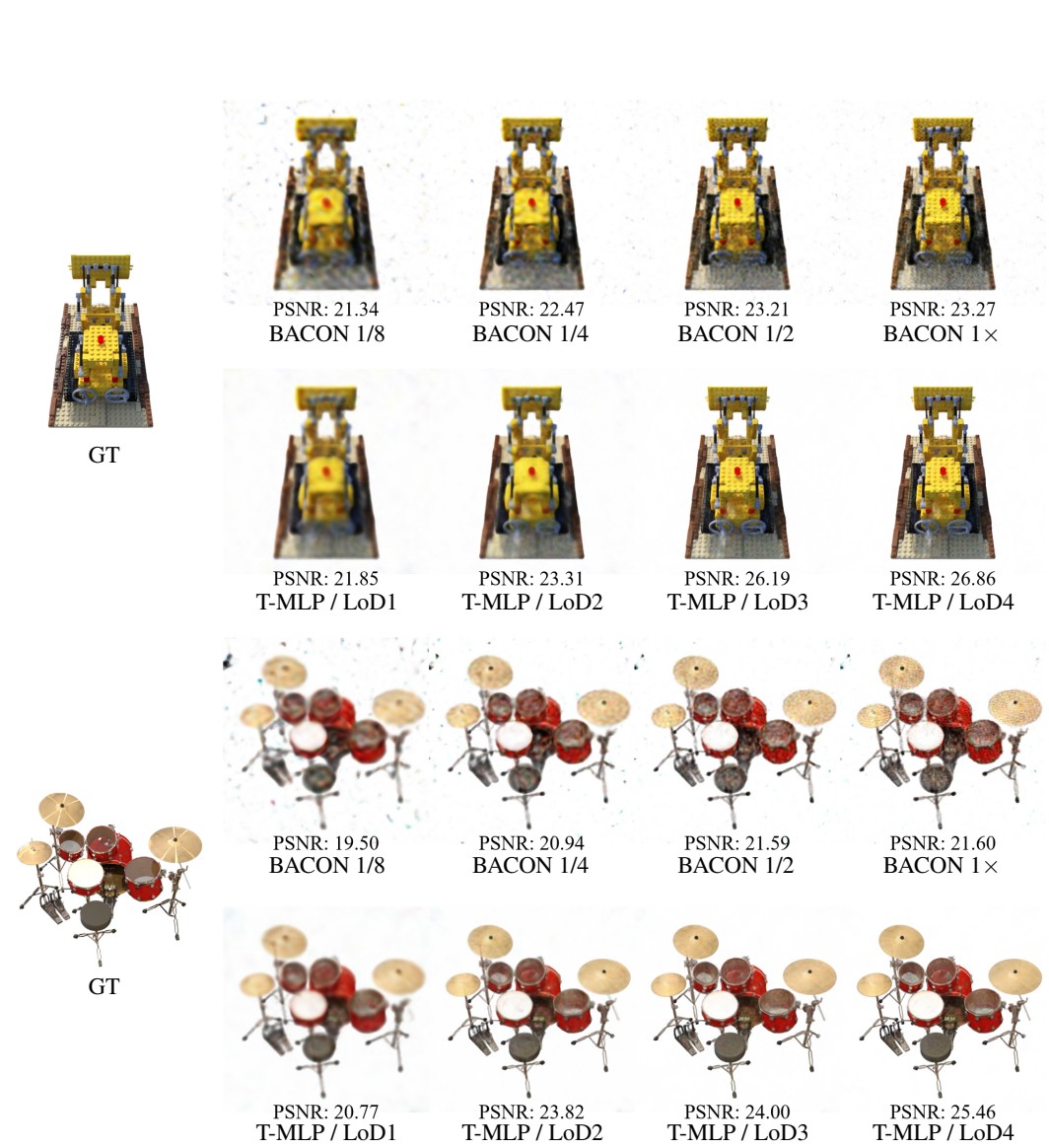

Figure A8: Visual comparisons of neural radiance field fitting under single-resolution image supervision.

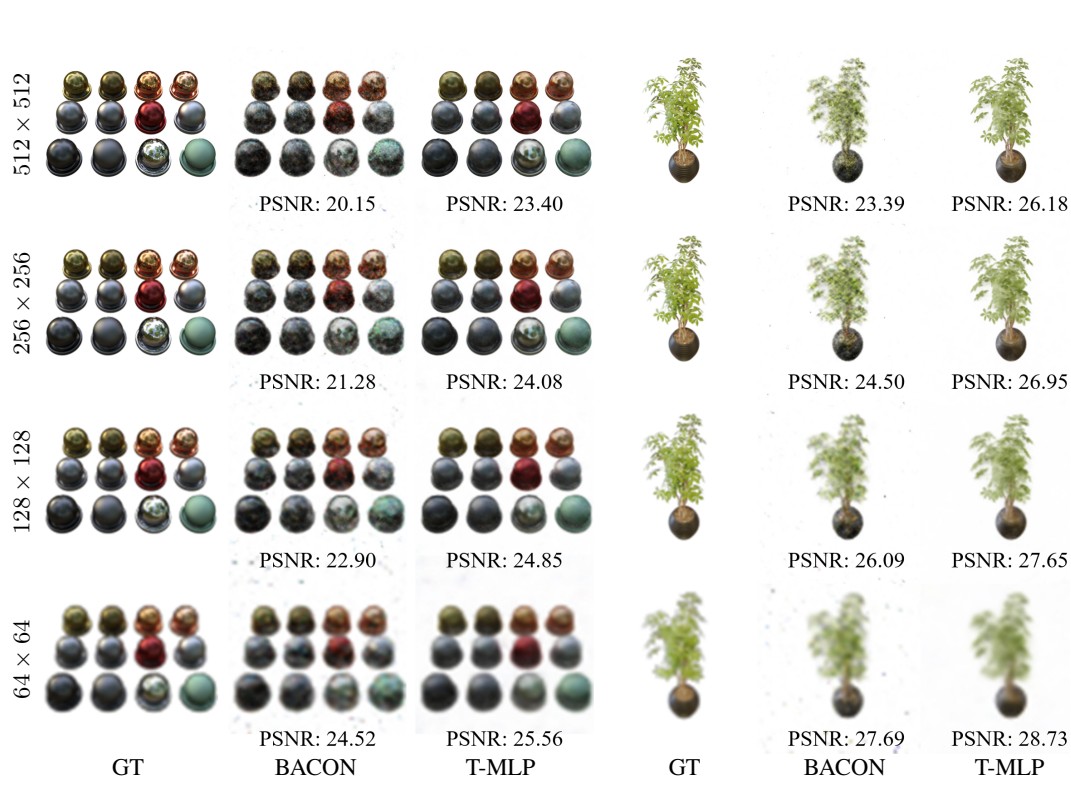

Figure A9: Visual comparisons of neural radiance field under multi-resolution image supervision. Note that the quantitative results are evaluated against ground-truth images at the corresponding resolutions.

## A.5 ABLATION STUDIES

### A.5.1 T-MLP VS MLP WITH RESIDUAL CONNECTION

We use an MLP with residual connections (He et al., 2016) to replicate the experiment described in Section 5.1 of the main paper, with results shown in Fig. A10. While residual connections improve gradient flow to early-layer hidden representations, the lack of explicit guidance prevents these early-layer hidden representations from producing satisfactory approximation of low-detail signals and from supporting LoD.

While both T-MLP and ResNet (He et al., 2016) employ the concept of residuals, their mechanisms are fundamentally different. ResNet uses a single output tail, requiring deeper layers to iteratively refine the hidden representation into a final form, which is then mapped to the output via this tail; thus, each hidden layer learns the residual between the current hidden representation and the ideal hidden representation. In contrast, T-MLP attaches multiple output tails, each iteratively predicting the residual between the current accumulated prediction and the ground truth, so that each hidden layer learns the hidden representation of the residual between the current prediction and the ground truth.

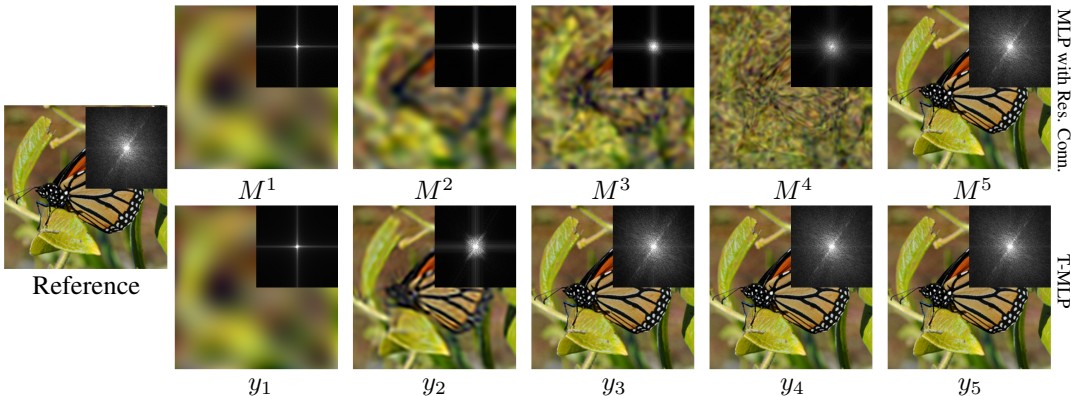

Figure A10: T-MLP VS MLP with Residual Connection. The image is from the DIV2K dataset (Agustsson & Timofte, 2017).

## A.6 LLM USAGE

Large Language Models (LLMs) were used solely as general-purpose writing assistants. They helped with grammar correction, phrasing suggestions, and formatting consistency. No part of the research design, methodology, or experimental results was generated by LLMs.

