# OpenReview forum: "T-MLP: Tailed Multi-Layer Perceptron for Level-of-Detail Signal Representation"
_ICLR.cc/2026/Conference — ICLR 2026 Conference Withdrawn Submission_

### Official Review · Reviewer_G91k · 2025-10-25

**Soundness:** 3
**Presentation:** 3
**Contribution:** 2
**Rating:** 4
**Confidence:** 4

**Summary:**

In this paper, the authors propose a Tailed Multi-Layer Perceptron (T-MLP) designed to achieve multiple levels of detail (LoD) across different signal representations. The core idea is to attach an output branch to each hidden layer, enabling intermediate feature extraction at various depths. The outputs from all layers are aggregated and jointly supervised to realize the desired LoD effects. The authors also conduct experimental comparisons to validate the effectiveness of the proposed T-MLP architecture.

**Strengths:**

+ Generally, the paper is well written which is easy to follow and understand.
+ The authors mostly follow the evaluation settings of existing methods to support their technical claims.

**Weaknesses:**

+ There is some related literature missing which also works on the multi-scale implicit representations. To name a few,
1) Neural Fourier Filter Bank, CVPR 2023
2) NeuRBF: A Neural Fields Representation with Adaptive Radial Basis Functions, ICCV 2023
3) FINER: Flexible spectral-bias tuning in Implicit NEural Representation by Variable-periodic Activation Functions, CVPR 2024

+ Some important baselines are missing such as Residual Multiplicative Filter Networks (NeurIPS 2022), InstantNGP (SIGGRAPH 2022), NFFB (CVPR 2023), WIRE (CVPR 2023) and more recent papers. These projects also focus on the multi-resolution modeling of neural implicit representations. I encourage the authors to add these baselines for more comprehensive comparisons.

+ Although it might be subjective, the technical novelty appears to be limited. Specifically, the statement "Our findings show that not...components of the signal" in L43-L46 was also explored in BACON (CVPR 2022), Residual Multiplicative Filter Networks (NeurIPS 2022) and NFFB (CVPR 2023). From what I can tell, the network architecture shown in Figure 1 looks a bit too similar to the architecture of Residual Multiplicative Filter Networks. The main differences might be that the proposed method builds upon linear layers instead of multiplicative filters.

+ The experimental results do not look promising. For example in Table 1, the scores are very similar to those of SIREN which was published five years ago. The visual improvements over SIREN are very minor as well.

+ As a pure MLP based network, the dense computation prevents achieving the real-time running efficiency.

+ There are no comprehensive quantitative comparison results for neural radiance fields which are a common testbed for neural fields.

**Questions:**

Please see the weakness part above.

---

### Official Review · Reviewer_Y8Pq · 2025-10-28

**Soundness:** 1
**Presentation:** 2
**Contribution:** 2
**Rating:** 2
**Confidence:** 5

**Summary:**

The paper introduces Tailed Multi-Layer Perceptron (T-MLP), a new architecture for level-of-detail (LoD) signal representation. By attaching output tails to each hidden layer, T-MLP enables multi-scale modeling. Each tail refines the residual from previous layers, allowing the network to represent signals at different LoD levels using only single-resolution supervision.

**Strengths:**

The paper presents a clear and well-motivated idea, and the writing is concise and easy to follow, making the technical contributions accessible. The proposed T-MLP architecture is conceptually simple yet effective, providing a straightforward way to achieve multi-scale or level-of-detail signal representation within an MLP framework. The experimental results convincingly demonstrate the effectiveness of the proposed method.

**Weaknesses:**

1. The paper extends the classic SIREN architecture by adding intermediate layers and a Polynomial Transformation, yet the necessity and contribution of these two components are not theoretically or experimentally justified. It remains unclear whether these modifications are essential for achieving the reported improvements.

2. In line 269, the description of “suitable affine transformations” lacks clarity. The paper should specify what these transformations refer to and why they are required to approximate the signal, as this seems to be a key assumption in the proposed framework.

3. In line 352, the rationale for the λ parameters is not clearly explained. The paper should discuss how these hyperparameters are chosen and balanced, particularly the reason for setting λ₁ = 0, which appears critical to the training objective.

4. In Table 3, the performance of LoD1 at 1024 resolution is inferior to BANF, which contradicts the overall claim of superiority. The authors should analyze and explain this inconsistency.

5. Regarding the Multiplicative Design, the improvement shown in Table 4 is very limited, suggesting that its contribution to overall performance is marginal. This is likely due to the additional parameters introduced by this design

**Questions:**

See weaknesses.

---

### Official Review · Reviewer_2ZcE · 2025-11-01

**Soundness:** 3
**Presentation:** 3
**Contribution:** 2
**Rating:** 4
**Confidence:** 4

**Summary:**

The paper addresses the problem of extending implicit neural representations (INRs) to support level-of-detail (LoD) signal representation. To this end, the authors modify the standard MLP-based INR architecture by attaching output branches (or tails) to each hidden layer, so that every intermediate layer produces a partial signal approximation, while subsequent layers predict and add residual refinements. This cumulative, coarse-to-fine formulation enables a single network to represent a signal at multiple levels of detail.

The method is compared against several existing INRs, including those designed for LoD modeling (e.g. NGLOD, BACON, BANF), mainly on neural surface representation and image fitting tasks. Across these benchmarks, the proposed approach achieves competitive or improved performance over the baselines. However, questions remain regarding its practical validation beyond surface-fitting experiments. It is not yet clear which real-world downstream applications the method can directly support. Demonstrating its integration with neural rendering pipelines (e.g. NeRF) or multi-view surface reconstruction methods (e.g. VolSDF [1] and follow-ups) would further strengthen the practical relevance of the approach.

**Strengths:**

**S1. Efficient and elegant residual design for INRs.** The proposed LoD supervision mechanism is conceptually simple, well-motivated, and easily integrable into modern INR architectures. Its ability to consistently improve surface reconstruction quality demonstrates the practicality and generality of the residual formulation, encouraging its adoption across a range of implicit representation tasks.

**Weaknesses:**

**W1. Limited evidence of practical relevance beyond controlled signal-fitting tasks.** The main concern lies in the unclear applicability of the proposed architecture to real-world tasks. While the method demonstrates convincing results on synthetic signal-fitting experiments (e.g., image and surface reconstruction from dense samples), it remains uncertain how effectively it transfers to practical scenarios. Integrating T-MLP into downstream applications—such as neural rendering (e.g., NeRF), where LoD-aware rendering could accelerate coarse-to-fine visualization, or multi-view neural surface reconstruction (e.g., VolSDF [1])—would provide a stronger validation of its practical utility and performance benefits.

**W2. Moderate conceptual novelty.** The proposed residual formulation and supervision scheme can be interpreted as a streamlined recombination of ideas already explored in earlier works such as BACON and BANF. Thus, while the resulting architecture is both elegant and effective, the conceptual advance beyond prior hierarchical or residual INR frameworks appears relatively incremental.

**W3. Limited ablation and insufficient analysis of computational trade-offs.** The paper would benefit from a more comprehensive ablation study, examining factors such as layer size, supervision weighting, and the number of tails. More importantly, the work lacks a clear analysis of the efficiency–accuracy trade-off that the proposed formulation aims to achieve. Beyond the quantitative summaries in Tables 2 and 4, presenting simple curves illustrating the relationship between number of parameters, inference time and reconstruction accuracy would substantially clarify the claimed computational advantages.

References

[1] Yariv, L., Gu, J., Kasten, Y., & Lipman, Y. (2021). Volume rendering of neural implicit surfaces. Advances in neural information processing systems, 34, 4805-4815.

**Questions:**

**Q1: Practical Relevance and Applicability.** Could the authors clarify how the proposed T-MLP architecture could be integrated into **practical downstream tasks** such as neural rendering (e.g., NeRF) or multi-view surface reconstruction (e.g., VolSDF)? Demonstrating or at least discussing such applications would help establish the method’s usefulness beyond controlled signal-fitting benchmarks.

**Q2: Ablation Studies and Efficiency–Accuracy Analysis.**. Would the authors consider extending their ablation analysis to include the effects of key design factors (e.g., layer width, number of tails, supervision weighting) and, importantly, provide a quantitative evaluation of the **efficiency–accuracy trade-off**? For instance, a plot showing reconstruction quality versus inference time across LoD levels could clarify the practical computational benefits.

**Q3: Novelty and Conceptual Contribution.** Finally, the residual formulation and supervision scheme seem related to ideas explored in earlier hierarchical or residual INRs (e.g., BACON, BANF). Could the authors comment on what they consider the **key conceptual innovation** of T-MLP relative to these prior frameworks?

---

### Official Review · Reviewer_DooM · 2025-11-03

**Soundness:** 2
**Presentation:** 3
**Contribution:** 2
**Rating:** 2
**Confidence:** 4

**Summary:**

This paper proposes T-MLP, a modified MLP architecture designed to enable Level-of-Detail (LoD) signal representation within the implicit neural representations (INRs). Instead of producing a single output at the final layer, T-MLP attaches output branches (“tails”) to each hidden layer. Each tail learns the residual between the accumulated prediction so far and the ground-truth signal, so the successive tails capture finer details. This design is claimed to allow:

(1) Multi-scale signal modeling without multi-resolution supervision,

(2) Progressive transmission, where early layers produce coarse outputs that can be refined with subsequent layers, and

(3) Improved optimization, since all hidden layers receive direct supervision.

Experiments across image fitting (DIV2K) and 3D shape SDF representation (Thingi10K, Stanford 3D scans) show improvements on signal representations compared with previous methods including SIREN, BACON, NGLOD, and BANF.

**Strengths:**

1. Attaching lightweight “tails” to each hidden layer provides a way to obtain multi-resolution outputs from a single MLP. It is easy to integrate into existing INR frameworks with such residual design.

2. The cumulative residual learning mechanism (Eq. 2-4) ensures early tails capture low-frequency components while deeper ones refine high-frequency details. This leads to interpretable layer-wise LoDs and improves training stability, also supporting scalable signal compression.

**Weaknesses:**

1. The idea of multi-output or residual-supervised layers is conceptually straightforward and reminiscent of cascade/residual networks. While well-executed, the step from “MLP with tails” to LoD representation is incremental rather than theoretically groundbreaking.

2. The empirical finding that deeper layers encode higher frequencies is reasonable, but the paper lacks formal frequency analysis or spectral decomposition to support this claim quantitatively. In fact, it is very straightforward that later layers capture high-frequency signals as it is trained as residual compensation.

3. The choice of layer-wise loss weights λ = (0, 0.5, 0.5, 0.5, 2.5) and multiplicative branch structure seems empirically tuned. It would help to include a stability study or guidelines for different tasks.

4. While the paper contrasts with NGLOD/BACON/BANF, a more detailed comparison to band-limited or coordinate-warped INRs (e.g., ACORN, Fourier volume methods) would strengthen positioning.

**Questions:**

1. How stable is training when the number of tails changes (e.g., 3 vs 7 layers). Residual structure design usually increases the training stability. So is it possible to make the INR network very deep?

2. Would integrating pruning (as mentioned in the discussion) reduce redundancy without harming LoD continuity?

3. While the authors claim to support scalable compression with the residual structures, how effective is it? The bits cost for deep network layers may not be worthy for the marginal PSNR improvement enhanced by deep layers.

---

### Note · Authors · 2026-01-22

I have read and agree with the venue's withdrawal policy on behalf of myself and my co-authors.